# Distribution of Glycerophospholipids in the Adult Human Lens

**DOI:** 10.3390/biom8040156

**Published:** 2018-11-22

**Authors:** Jo Ann Seng, Jessica R. Nealon, Stephen J. Blanksby, Todd W. Mitchell

**Affiliations:** 1School of Chemistry, Faculty of Science, Medicine and Health, University of Wollongong, Wollongong, NSW 2522, Australia; joannseng@gmail.com; 2School of Medicine, Faculty of Science, Medicine and Health, University of Wollongong, Wollongong, NSW 2522, Australia; jnealon@uow.edu.au; 3Illawarra Health and Medical Research Institute, Wollongong, NSW 2522, Australia; 4Central Analytical Research Facility, Institute for Future Environments, Queensland University of Technology, Brisbane, QLD 4000, Australia; stephen.blanksby@qut.edu.au

**Keywords:** imaging mass spectrometry, lipidomics, phosphatidic acid, phosphatidylserine

## Abstract

In humans, the age of fibre cells differs across the ocular lens, ranging from those formed before birth in the core of the lens to those formed just prior to death in the outer cortex. The distribution of glycerophospholipids in the adult human lens should reflect this range; however, limited data currently exists to confirm this hypothesis. Accordingly, this study aimed to determine the distribution of glycerophospholipids in adult human lens using mass spectrometry imaging. To achieve this, 20-µm thick slices of two human lenses, aged 51 and 67 were analysed by matrix-assisted laser desorption ionisation imaging mass spectrometry. The data clearly indicate that intact glycerophospholipids such as phosphatidylethanolamine, phosphatidylserine, and phosphatidic acid are mainly present in the outer cortex region, corresponding to the youngest fibre cells, while lyso-phosphatidylethanolamine, likely produced by the degradation of phosphatidylethanolamine, is present in the nucleus (older fibre cells). This study adds further evidence to the relationship between fibre cell age and glycerophospholipid composition.

## 1. Introduction

The lens is an ocular tissue that grows throughout the life span of an organism with newly differentiated fibre cells laid down upon older cells that are present from birth [1]. There is no protein [2] or lipid [3] turnover in the lens, thus structural and enzymatic proteins and lipids that are present at birth remain for the lifetime of the individual. In addition, upon fibre cell differentiation, all intracellular organelles are degraded [4]. Therefore, the lens that is present from birth (known as the nucleus) is composed of plasma membranes packed with a protein dense cytosol [5]. Human cellular membranes are comprised of a lipid bilayer containing phospholipids and integral membrane proteins. A typical membrane consists mainly of phospholipids and glycolipids of different classes, headgroups and fatty acyl chains. Membrane fluidity is partly regulated by the degree of unsaturation of the fatty acyl chain [5]. Unlike a typical membrane, the adult human lens membrane is one of the most saturated and ordered membranes [6]. It contains a high level of cholesterol and saturated dihydrosphingolipids [7]. Also present in lens membranes are glycerophospholipids (GPL), which consist of a three-carbon glycerol backbone where two fatty acids attach at the *sn*-1 and *sn*-2 positions and a phosphate headgroup attached at the *sn*-3 position [7]. There are six main classes of glycerophospholipids present in the human lens: phosphatidylcholine (PC), phosphatidylethanolamine (PE), phosphatidylserine (PS), phosphatidylinositol (PI), phosphatidylglycerol (PG), and phosphatidic acid (PA) [7,8].

Phosphatidylcholine is the most abundant phospholipid in animals lenses, accounting for 31–46% of total phospholipids in the rat, chicken, sheep, cow, and pig [7]. In contrast, PC in the human lens accounts for only 11% of the total phospholipidome, while the most abundant GPL in the human lens is phosphatidylethanolamine (15%) [7,9,10]. In the human lens, two thirds of phosphatidylethanolamine and phosphatidylserine carry the unusual 1-*O*-alkyl ether-linkage; contributing more than 50% and 65% of the total PE and PS content, respectively [7,9]. These alkyl ether-linked phospholipids are associated with lens transparency, as the inhibition of their synthesis results in cataract in mice [11]. Lyso-derivatives of glycerophospholipids are also observed, which possess only a single fatty acyl chain linked to the glycerol backbone [7,9].

Although glycerophospholipids have previously been identified and quantified in the human lens [7,10,12,13]**,** there are limited data on their distribution within the tissue. Ellis et al. imaged the human lens using desorption-electrospray ionisation mass spectrometry (DESI-MS) and showed the distribution of three glycerophospholipids, namely PE O-18:1_18:1, LPE O-18:1, and PS O-18:1e_18:1 [14]. The study did not include other classes of glycerophospholipids such as PA that have been recently identified [7,8]. Previous imaging studies using matrix-assisted laser desorption ionisation mass spectrometry (MALDI-MS) on human lens tissue have focussed on more abundant sphingolipids [15]. In this study, we aimed to investigate the distribution of glycerophospholipids in the human lens using MALDI-MS, exploiting the higher spatial resolution compared to DESI-MS and a higher resolving power mass analyser to provide greater confidence in glycerophospholipid assignment.

## 2. Materials and Methods

### 2.1. Materials

The matrix 2,5-diaminonapthalene (DAN) was purchased from Sigma−Aldrich (Castle Hill, Australia). Glass microscope slides for tissue mounting were purchased from Proscitech (Townsville, Australia). The optimal cutting temperature (OCT) compound used to adhere the lenses to the glass slides was purchased from Sakura Finetek (Torrance, CA, USA).

### 2.2. Lenses

Human lenses were obtained from eyes donated to the New South Wales Lions Eye Bank at the Sydney Eye Hospital (Sydney, Australia) within 2–6 h of death, and were stored immediately at −80 °C until required. All work was approved by the human research ethics committee at the University of Wollongong (HE 13/401).

### 2.3. Matrix-Assisted Laser Desorption Ionisation Mass Spectrometry

For MALDI-MS analysis, lenses were sliced in the transverse plane to produce 20-µm thick slices that were then thaw-mounted onto a glass slide and dried in a desiccator for 10 min prior to sublimation. Matrix (DAN) was sublimed at 30 mTorr and 112 °C for 15 min. Samples were analysed by an LTQ-Orbitrap XL fitted with an intermediate pressure MALDI source (Thermo Scientific, Bremen, Germany). Mass spectra were acquired in negative ion mode with a laser raster step size of 100 µm and an average of 50 laser shots per spectrum. The resolving power at *m*/*z* 400 was 33,000 and the mass range acquired was *m*/*z* 400–2000. Lipids were identified based on accurate mass, with an error (Δ) of ±3 ppm in combination with previously published lipid identifications in the human lens [7,10,16,17]. Mass spectrometry imaging data were visualised by ImageQuest 1.0 (Thermo Scientific, Bremen, Germany). Images of the selected *m*/*z* values were normalized to the total ion current (TIC). Lipid nomenclature follows the recommendations of Liebisch et al. [18].

## 3. Results and Discussion

### 3.1. Phosphatidylethanolamine

The most abundant PEs in the human lens contain 1-*O*-alkyl ether linkages [7]. Figure 1 shows the distribution of lysophosphatidylethanolamine (LPE) and PE species in a 51-year-old and 67-year-old human lens, respectively. A total of 15 species of PE and four species of LPE were detected across both human lenses (see Appendix A).

Figure 1 demonstrates that different LPE species were present in both the cortex and the nuclear regions in both lenses. Most LPE species that were detected had a higher relative abundance in the nucleus compared to the cortex. This observation was more pronounced in the 51-year-old lens compared to the 67-year-old lens. As expected, based on data from previous quantitative studies, the signal intensity of LPE P-18:1 in the 51-year-old lens was lower than the other three species detected, and its distribution was dispersed mostly throughout the nucleus. 

The mass spectral images revealed PE lipids located in an annular ring around the outer region of the cortex (Figure 1). This is in agreement previous with observations using DESI [14]. The PE O-18:1_18:1 was the most abundant PE in the human lens; contributing approximately 40% to the total PE [7]. The PE P-18:1_18:1, like its lyso analogue, displayed a lower signal intensity compared to other PE species. There was no PE O-18:0_18:1 detected in the 67-year-old lens. The annular distribution of PE revealed in the images in Figure 1 is in agreement with quantitative analysis of dissected lenses indicating that PE is approximately 1000-fold more abundant in the outer region compared to the nuclear or barrier regions of the lens [14].

Interestingly, PE O-16:0_18:1 and PE O-18:0_18:1 and their respective LPE analogues showed a complementary distribution, i.e. PE were detected at higher abundances in the outer region and LPE were detected at higher abundances in the nucleus. We have previously obtained similar results using DESI imaging, where PE was detected around the lens perimeter whilst LPE was distributed in an annular pattern in the cortex, inside that of PE [14]. Unlike the current study, Ellis et al. [14] did not observe LPE in the nuclear region using DESI imaging. The older lens nucleus contain more compact fibre cells which causes its consistency to be more rigid as compared to the cortex [19]. It is therefore possible that the spray solvent used in DESI was not able to fully extract and desorb analytes in the compact nuclear region, resulting in a suppressed signal intensity of LPE detected in the nucleus.

### 3.2. Phosphatidic Acid and Phosphatidylserine

The distribution of representative PA and PS lipids are shown in Figure 2. Both lipid classes showed a similar distribution despite differences in absolute signal intensity. Figure 2 shows the distribution of PA in the human lens with the ether-linked PA O-16:0_18:1 distributed in a hollow ring, spatially consistent with the outer region of the cortex in the 51-year-old lens. The same species showed a similar distribution in the 67-year-old lens albeit with lower overall intensity than was observed in the 51-year-old lens. In contrast, PA O-16:0_18:1 showed a more homogeneous distribution in the cortex, with a higher intensity around the edge of the 51-year-old lens and a lower intensity in the nucleus. In the 67-year-old lens, the distribution of this PA lipid was similar; however, the difference in signal intensity was not as great between the outer edge and the nucleus as compared to the 51-year-old lens. Phosphatidic acid is an intermediate in the biosynthesis of a wide-range of phospholipids in mammals, thus the localisation of these lipids in the outer region of the human lens could be suggestive of active lipid metabolism [20].

The distribution of seven species of PS in the human lens were obtained using this method and are shown in Appendix A. The distribution of two abundant species, PS O-16:0_18:1 and PS O-18:1_18:1 [7], is shown in Figure 2. Both PS species showed an annular distribution around the outer edge of the lens in the 51- and 67-year-old lens, and are broadly consistent with prior observations made using DESI imaging [14].

## 4. Conclusions

This study demonstrates that MALDI-MS is a powerful tool that can be used to observe the localisation of different glycerophospholipids in the human lens. Glycerophospholipids present in the adult lens were primarily detected in the cortex except for LPE. This agrees with previous quantitative data, where glycerophospholipids in the nucleus were depleted in the adult lens after the age of 40 [12]. Lysophosphatidylethanolamine showed a higher relative abundance in the nucleus of the lens in comparison to its PE analogue, which agrees with work indicating that LPE concentration is 4–5 fold higher than PE in the adult lens nucleus [12,21]. The ability to desorb and ionise LPE from the lens nucleus—a highly compacted tissue—may suggest an advantage of the MALDI approach for future lens imaging studies.

Overall, the annular distribution of intact glycerophospholipids revealed here by MALDI imaging are consistent with greater susceptibility of these molecules to oxidative and hydrolytic degradation over time, in contrast with more robust sphingolipids [22]. As there is no lipid turnover in the inner parts of the lens to replenish degraded GPLs [3], a diminishing concentration from the outer to the core is expected with age and validated by our observations [12]. Overall the decrease in GPL will result in a concomitant increase in the ratio of sphingolipids to GPLs, which may decrease diffusion through the tissue and protect against further oxidation and degradation [22].

The spatial distribution data provided by imaging mass spectrometry reveals the location of different lipids in different regions of the adult human lens. These data combined with other lipid imaging studies can be used to complement the growing understanding of lens protein distributions [23,24] and improve our understanding of the interaction between lens proteins and lipids that contribute to lens cellular function. Further studies to determine the abundance of these lipids, or the changes of their distribution with age are required in order to understand the potential role of these lipids in different regions of the lens.

## Figures and Tables

**Figure 1 biomolecules-08-00156-f001:**
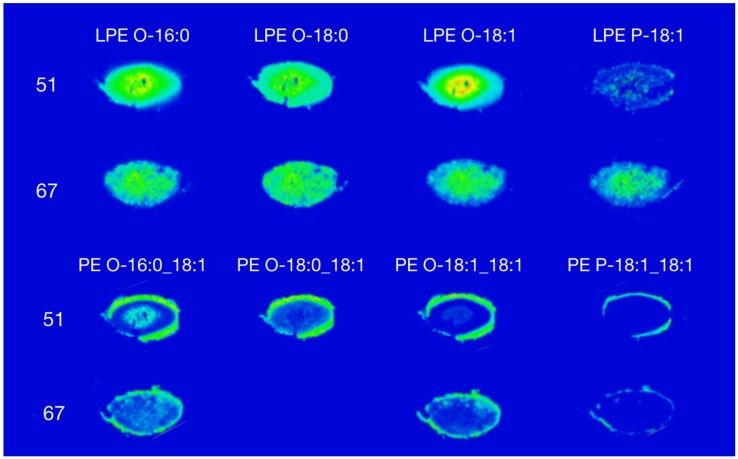
Distribution of lysophosphatidylethanolamine (LPE) and the most abundant phosphatidylethanolamines (PEs) in a 51-year-old and 67-year-old human lens.

**Figure 2 biomolecules-08-00156-f002:**
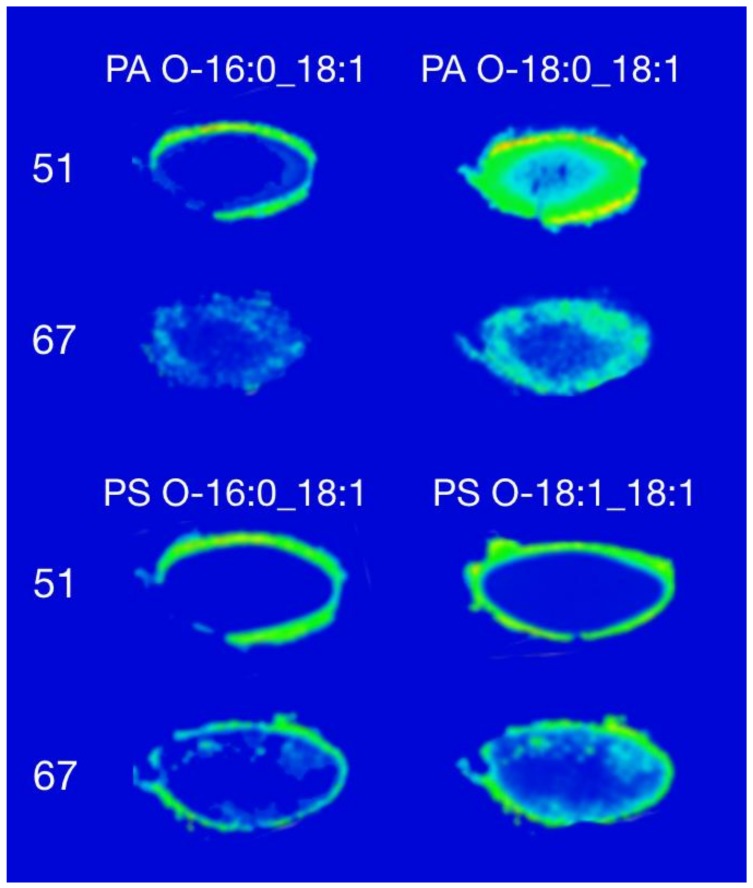
Distribution images of PA and PS in a 51-year-old and 67-year-old human lens. All lipids were detected with [M − H]^−^ in negative ion mode. PA: phosphatidic acid, PS: phosphatidylserine.

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
