# Peer review of "Distribution of Glycerophospholipids in the Adult Human Lens"

_biomolecules, 2018, doi:10.3390/biom8040156_

Round 1

Reviewer 1 Report

The distribution of glycerophospholipids in human lenses was observed using MALDI-MS. The data provides evidence for age related changes in the distribution of phospholipids. The study was performed in an experienced, well respected laboratory. The manuscript was well written with a wonderful list of citations. My only very minor suggestion is that the authors may like to provide a short insight into their choice of matrix. If you think of any factors that could influence the intensity of the signal you might want to include them if appropriate (quenching moieties, fiber cell density and matrix distribution.) I look forward to seeing the study published.

Reviewer 2 Report

This paper is devoted to the study of spatial distribution of glycerophospholipids in the human lens. The measurements were performed with the use of MALDI-MS imaging – a relatively novel method to visualize the distribution of proteins, peptides and metabolites in slices of biological samples. The most important finding of the work is the observation that while the intact lycerophospholipids are mainly present in the outer cortex, the lens nucleus is enriched with degraded phospholipids (lyso-phosphatidylethanolamine). Very likely, that demonstrates the process of age-related phospholipid degradation in the human lens. The results of the work may make an essential contribution into understanding of the lens aging and cataractogenesis. Regretfully, the measurements were performed for two lenses of similar ages only; it would be interesting to study the age-related changes in glycerophospholipid distribution, as well as the cataract-related changes. Hopefully, the authors will do that study in the future. The work was done professionally, the paper is well written and the arguments are generally clear. In my opinion, the manuscript can be published in Biomolecules.